applied mathematics

safety evaluation, underground gas storage, matter-element extension method, vague set, entropy

**Author for correspondence:**
Aorui Bi
e-mail: bar_wayne@xauat.edu.cn

# Comprehensive weighted matter-element extension method for the safety evaluation of underground gas storage

## Aorui Bi, Zhengshan Luo, Yulei Kong and Lexin Zhao

School of Management, Xi'an University of Architecture and Technology, Xi'an, Shaanxi 710055, People's Republic of China

AB, 0000-0002-0467-5545; ZL, 0000-0001-7950-898X; YK, 0000-0002-3904-5906; LZ, 0000-0001-7662-6315

This study focuses on a safety evaluation method for underground gas storage. Gas storage is usually constructed underground in complex environments, and the service life of such facilities is limited. To ensure the secure and long-term operation of gas storage facilities, safety evaluation has become the focus of management. The present paper provides an effective method for safety evaluation. An index system was established as the foundation of the analysis for this evaluation, and the matter-element extension method was applied to obtain a quantitative evaluation result. For the weight values of each index in the matter-element extension method, this paper presents a comprehensive weight computation method based on vague sets and entropy. By application of this method, the safety level of a gas storage facility in the Jintan salt mines (in Jiangsu, China) was calculated, and the evaluation result was 4.6433, which meant the safety level was V and the underground gas storage was slightly at risk. It indicated that the influence on the overall safety and tightness of this gas storage could be ignored in the operation process, but the frequency of regular monitoring should be increased. The defective indexes were also obtained, such as salt rock cohesion, the roof thickness, the volume contraction ratio, the interlayer content, the height of the casing shoe and the adjacent cavity pressure difference, which need to be monitored and modified. This paper evaluated the safety of the underground gas storage from a unique perspective. It is expected that the results of this research will contribute to the maintenance and operational decisions, and provide a reference for management in the energy industry.

# 1. Introduction

The consumption of natural gas may not be constant but fluctuate seasonally, which is not compatible with the extraction process. To solve this problem, surplus natural gas is usually stored in facilities. Additionally, the facilities should be able to store large amounts of natural gas safely and adapt to the pressure changes caused by cycle of storage or extraction. Therefore, as a good choice for natural gas storage, underground storage facilities are safer than other storage facilities. However, when underground gas storage facilities undergo damage (e.g. gas leakage, cavity damage, ground subsidence, etc.), the resulting catastrophic accidents will be very serious, especially in some countries with high population density; for instance, most gas storage facilities in China were built near cities with millions of inhabitants and well-developed infrastructure. The damage to storage will affect millions of people's lives and working conditions. Therefore, safety is among the most important criteria during the operation of underground storage.

The USA and some developed countries have made a plan for the strategic reserve of gas in underground storage since the 1970s and have also studied the safety of storage methods. The studies on safety are mainly focused on the mechanical properties of the geological storage medium and the establishment of constitutive models; these studies provide a scientific basis for safety evaluation. Dreyer [1] studied the mechanical properties of salt rock under different pressures. Regarding creep properties, there have been many studies that have obtained different forms of elastic–plastic constitutive models [2–7]. Based on the elastic–plastic constitutive model, creep damage models have been formed and used to study permeability, damage and self-healing of salt rock that can be used for underground storage [8–12]. With the development of computer technology, computer-aided engineering software could be used to study the nonlinear correlation between time and the properties of a geological medium [13–16]. Through more-detailed research on salt caverns, it was found that cavity parameters also affect safety. Field testing has suggested that the rheological properties of salt rock have led to a volume loss in the French TE02 gas storage facility in excess of 30% during its 10-year operation [17] and that the volume loss rate increases with increasing burial depth [18]. However, all of the above-mentioned properties are passive and usually influenced by mechanical properties. Many experiments have found that mechanical properties such as the shear stress, pore pressure and Lode angle will change the rock creep, cavity volume and other properties [19–21].

The existing studies have mostly focused on the impacts of a single index; until now, there was no unified global standard for safety evaluation of underground gas storage. To make a study on safety more valuable for sustainable management, comprehensive safety evaluation is needed. In this paper, an index system for safety was established, including three secondary index sets of mechanical properties, cavity parameters and operation parameters, a total of 16 safety indexes. The evaluation levels were set up from safety to unsafety and divided the values of each index by referring the levels. We defined vague values for each evaluation level and gave a vague value for each index by different experts, and then calculated the subjective weights. The objective weights were calculated using the entropy weight method. By minimizing the square error, we obtained the comprehensive weights. Finally, combined with the matter-element extension method, the safety level of the underground gas storage was obtained. According to the safety level, the current safety status of the storage could be judged, and based on the status, the maintenance and operation decisions could be made to ensure that the storage formed a more reliable and long-lasting operation.

The present paper is organized as follows: Section 3 introduces the extension model. Section 4 describes the design of the computational method of index weighting in the extension model. Section 5 provides an example to validate the model, then discusses and analyses the evaluation results. Finally, §6 provides the conclusion of this paper.

# 2. Matter-element extension evaluation method

A matter-element extension evaluation method is developed on the basis of a Cantor set and fuzzy set, to solve the evaluation problem in both qualitative and quantitative ways [22,23], and the method has been applied in many fields [24–26]. A matter-element is the basic unit of this method and is a logical structure described by the names, the characteristics and the quantities of an object. The basic evaluation processes are described as follows:

(i) Define the matter-element. In matter-element theory, there are three kinds of matter-elements, namely, the classical domain, joint domain and evaluation objects. The classical domain is defined as

$$R_p(q) = (U_q, C_q, V_q),$$ (2.1)

where $U_q$ is the level, $C_q$ is a feature set under $U_q$ and $V_q$ is the value range of $C_q$. In this paper, $U_q$ represents the evaluation level, and $C_q$ represents the evaluation index of gas storage.

The joint domain is defined as

$$R_u(p) = (U_u, C_u, V_u),\tag{2.2}$$

where $U_u$ is the set of objects under all levels, $C_u$ is the index collection of $U_u$ and $V_u$ is the value range of $C_u$.

The matter-element to be evaluated is defined as

$$R_p = (N_p, C_p, V_p),\tag{2.3}$$

where $N_p$ is the index set, $C_p$ is the index in $N_p$ and $V_p$ is the value of $C_p$.

(ii) Establish the correlation function and calculate its value. The correlation function is used to characterize the extension set, the set that is used to describe the transformation from the objects that do not have certain properties to other objects that have those properties. The correlation function value of the index under each level can be calculated by the following:

If $\rho[v_{ij}, v_{iuj}] - \rho[v_{ij}, v_{ipj}] = 0$

$$K_p(c_{ij}) = \frac{\rho[v_{ij}, v_{ipj}]}{\rho[v_{ij}, v_{iuj}] - \rho[v_{ij}, v_{ipj}]}.\tag{2.4}$$

If $\rho[v_{ij}, v_{iuj}] - \rho[v_{ij}, v_{ipj}] \neq 0$

$$K_p(c_{ij}) = -\rho[v_{ij}, v_{ipj}] - 1,\tag{2.5}$$

$$\rho[v_{ij}, v_{ipj}] = \left| v_{ij} - \frac{a_{ipj} + b_{ipj}}{2} \right| - \frac{b_{ipj} - a_{ipj}}{2}\tag{2.6}$$

and

$$\rho[v_{ij}, v_{iuj}] = \left| v_{ij} - \frac{a_{iuj} + b_{iuj}}{2} \right| - \frac{b_{iuj} - a_{iuj}}{2},\tag{2.7}$$

where $K_p(c_{ij})$ represents the correlation value, $\rho[v_{ij}, v_{iuj}]$ is the distance between $v_{ij}$ and $v_{iuj}$, $v_{ij}$ is the value of the $j$th index in the $i$th index set of the matter-element, $v_{ipj}$ is the $j$th index value range under the $q$th level in the $i$th index set of the classical domain, $v_{iuj}$ is the $j$th index value range in the $i$th index set of the joint domain and $\langle a_{ipj}, b_{ipj} \rangle$ and $\langle a_{iuj}, b_{iuj} \rangle$ are the value ranges of $v_{ipj}$ and $v_{iuj}$, respectively.

The correlation degree of the matter-element to be evaluated at all levels is calculated by

$$K_s = \sum w_{ij} K_p(c_{ij}),\tag{2.8}$$

where $K_s$ is the comprehensive correlation value, $w_{ij}$ is the weight value of the index and the sum of the weight values must be equal to 1.

(iii) Calculate the evaluation level eigenvalue. The level to which the maximum value of $K_s$ belongs is the final level of safety. The level eigenvalue of gas storage can be calculated by

$$s' = \frac{\sum_{s=1}^{i} sK'}{\sum_{s=1}^{i} K'_s},\tag{2.9}$$

with

$$K'_s = \frac{K_s - \min[K_s]}{\max[K_s] - \max[K_s]},\tag{2.10}$$

where $s'$ is the final evaluation result.

# 3. Design of the comprehensive weighting calculation method

For the matter-element extension evaluation method, the key step is to obtain the accurate index weight. Generally, to reflect the importance of indexes for the safety, each index need to be quantified by weight value, so how to determine the weights is the key step in the evaluation method. At present, the weighting calculation methods are usually divided into subjective and objective forms.

The subjective weighting method is based on the knowledge or experiences of decision-makers, by evaluating and quantifying the relevance or importance of the index to safety for obtaining the weights. The subjective weighting method can emphasize the intentions of decision-makers, but

sometimes wrong evaluation will lead to certain risk of misjudgement. Also, subjective weight reflects the preference of decision-makers and is targeted, but an arbitrary weight with poor transparency creates a certain risk of misjudgement.

The objective weight is calculated by the practical data with a strong mathematical basis. The objective weight depends on the number of data samples, and the calculation method is usually complex; it sometimes cannot reflect the importance of different indexes, and means the importance of the index is contrary to reality.

This paper applies the vague set theory and the entropy method to calculate the subjective and objective weights, respectively, and makes full use of objective data to carry out the comprehensive weighting of evaluation indexes on the premise of considering the subjective experience.

## 3.1. Subjective weight design based on vague sets

For subjective weight, the evaluation information of indexes cannot be represented by an exact value; there must be some uncertainty in the expression of these indexes. However, the actual situation and experience show that the evaluation information must have upper and lower limits for different levels of standards, which means there is a value interval for each index at each level. Therefore, this paper uses a vague value in the form of the numerical interval to express the evaluation information of the indexes. The vague value takes the negative and uncertain effects into consideration and is conducive to the accurate analysis of uncertain information. This approach has been widely used in the field of decision management, so this paper applies the vague set theory to design the subjective weight. The basic theory of vague sets is shown in appendix A.

The specific steps are as follows:

(i) The vague value of the index can be constructed by experts' evaluation, and the evaluation matrix for safety is described by the below equation

$$A = \begin{pmatrix} a_{11} & a_{12} & a_{13} & \dots & a_{1m} \\ a_{21} & a_{22} & a_{23} & \dots & a_{2m} \\ \dots & \dots & \dots & \dots & \dots \\ a_{n1} & a_{n2} & a_{n3} & \dots & a_{nm} \end{pmatrix}, \tag{3.1}$$

where $a_{nm} = [t_{nm}, 1 - f_{nm}]$ is the vague value of the index.

(ii) The consistency matrix is represented in the below equation, which is constructed by the similarity formula in definition A.2 (see appendix A):

$$M^k = \begin{pmatrix} M_{11}^k & M_{12}^k & M_{13}^k & \dots & M_{1n}^k \\ M_{21}^k & M_{22}^k & M_{23}^k & \dots & M_{2n}^k \\ \dots & \dots & \dots & \dots & \dots \\ M_{n1}^k & M_{n2}^k & M_{n3}^k & \dots & M_{nn}^k \end{pmatrix} \tag{3.2}$$

where $M_{ij}^k$ is the similarity value between vague values evaluated by different experts. Therefore, $M^k$ is the measure of the similarity of all the experts' evaluation to the index. If the similarity value is closer to 1, the experts have more consistent opinions, and if the similarity value is closer to 0, the experts' opinions are inconsistent.

(iii) The average similarity value for the index is represented by

$$V_i^k = \frac{1}{n} \sum_{j=1}^{n} M_{ij}^k. \tag{3.3}$$

The relative consistency values of each expert to every index are mathematically represented by

$$b_{ik} = \frac{V_i^k}{\sum_{i=1}^{n} V_i^k}. \tag{3.4}$$

(iv) Therefore, the relative consistency measure matrix of indexes evaluated by experts is represented by

$$B = \begin{pmatrix} b_{11} & b_{12} & b_{13} & \dots & b_{1m} \\ b_{21} & b_{22} & b_{23} & \dots & b_{2m} \\ \dots & \dots & \dots & \dots & \dots \\ b_{n1} & b_{n2} & b_{n3} & \dots & b_{nm} \end{pmatrix}, \tag{3.5}$$

where $B$ can be interpreted as the information representation of each expert's preference, and the indexes of the preference information for all the experts are defined as follows:

$$a_k = \sum_{i=1}^{n} B \otimes a_{ik} = \left[ \sum_{i=1}^{n} b_{ik} t_{ik}, 1 - \sum_{i=1}^{n} b_{ik} f_{ik} \right] \tag{3.6}$$

where $a_k$ is the evaluation vague value of all experts.

(v) Finally, according to $a_k$, the weight is defined as

$$w_s^k = t_k + \frac{(1 - t_k - f_k)}{2}, \tag{3.7}$$

where $w_s^k$ is the subjective weight and includes two parts: agreement and uncertainty. For the part of uncertainty, this paper defines half of it as agreement and the other half as disapproval. $t_k$ is the truth-membership degree of the vague value, whereas $f_k$ is the false-membership degree of the vague value, where $t_k$ and $f_k$ are represented in the below equations

$$t_k = \sum_{i=1}^{n} b_{ik} t_{ik} \tag{3.8}$$

and

$$f_k = \sum_{i=1}^{n} b_{ik} f_{ik}. \tag{3.9}$$

## 3.2. Objective weight based on the entropy method

Entropy is used to describe the uncertainty of information or data in information theory, i.e. the effective information after eliminating the redundant information. The smaller the entropy of information, the more content it contains; conversely, the larger the entropy of information, the less content it contains. To calculate the weight value based on entropy, the index information availability is used to measure the index. The steps of the entropy method are as follows:

(i) Define the index value matrix $Y = [y_{ij}]_{m \times n}$, where $y_{ij}$ is the value samples of indexes. Normalize $Y$ and obtain the matrix $Z$: $Z = [z_{ij}]_{m \times n}$, where $z_{ij}$ is the normalized value of $y_{ij}$. There are two evaluation forms for the index: for one form, the greater the index value is, the better the index, and for the other, the smaller the index value is, the better the index. The normalized equations are as follows:

$$z_{ij} = \frac{y_{ij} - y_{\min}}{y_{\max} - y_{\min}} \tag{3.10}$$

and

$$z_{ij} = \frac{y_{\max} - y_{ij}}{y_{\max} - y_{\min}}. \tag{3.11}$$

(ii) The entropy value is calculated as follows:

$$H_i = -\frac{\sum_{j=1}^{n} f_{ij} \ln f_{ij}}{\ln n}, \tag{3.12}$$

where $H_i$ is the entropy value, and $f_{ij}$ is given by

$$f_{ij} = \frac{z_{ij}}{\sum_{j=1}^{n} z_{ij}}. \tag{3.13}$$

To make $\ln f_{ij}$ meaningful, if $f_{ij} = 0$, set $\ln f_{ij} = 0$.

(iii) The index objective weight is obtained by

$$w_o^i = \frac{1 - H_i}{m - \sum_{i=1}^{m} H_i}, \tag{3.14}$$

for $0 \leq w_i \, o \leq 1$ and $\sum_{i=1}^{m} w_o^i = 1$.

## 3.3. Comprehensive weight methods

Weight designed based on vague sets shows the subjective opinions of experts, but all the evaluation results are highly subjective. The entropy method has a strong mathematical basis, but it does not reflect the intentions of the decision-makers. Therefore, the subjective and objective weights calculation methods have different disadvantages; there are deviations between the subjective/objective weights and optimal weights. To minimize the deviation, the reasonable approach is to combine the subjective weight and the objective weight; the deviation is defined as

$$\min \text{err} = \theta \sum_{i=1}^{n} [\bar{w}(w_i - w_s)]^2 + (1 - \theta) \sum_{i=1}^{n} [\bar{w}(w_i - w_o)]^2, \tag{3.15}$$

with

$$\bar{w} = \frac{(w_s + w_o)}{2} \tag{3.16}$$

and

$$\sum_{i=1}^{n} w_i = 1, \tag{3.17}$$

where $w_i$ is the optimal weight, err is the deviation between the comprehensive weight and $w$, $w_s$ is the subjective weight, $w_o$ is the objective weight, $\theta$ is the reliability of the subjective weight, and $1 - \theta$ is the reliability of the objective weight.

$w$ is evaluated by the equation of maximum-likelihood estimate

$$L(w_i, \lambda) = \theta \sum_{i=1}^{n} [\bar{w}(w_i - w_s)]^2 + (1 - \theta) \sum_{i=1}^{n} [\bar{w}(w_i - w_o)]^2 - 2\lambda \left( \sum_{i=1}^{n} w_i - 1 \right). \tag{3.18}$$

Take partial derivatives with respect to $w_i$ and $\lambda$, separately, and make the results equal to 0

$$\frac{\partial L}{\partial w_i} = 2\theta\bar{w} \sum_{i=1}^{n} [\bar{w}(w_i - w_s)] + 2(1 - \theta)\bar{w} \sum_{i=1}^{n} [\bar{w}(w_i - w_o)] - 2\lambda = 0 \tag{3.19}$$

and

$$\frac{\partial L}{\partial \lambda} = \sum_{i=1}^{n} w_i - 1 = 0. \tag{3.20}$$

Based on equations (3.19) and (3.20), the homogeneous equations are solved as

$$w_k = \left[ \frac{1 + \sum_{i=1}^{n} (b_k - b_i)/a_i}{a_k \sum_{i=1}^{n} 1/a_i} \right], \tag{3.21}$$

with

$$a_i = \sum_{i=1}^{n} \bar{w}_i^2, \quad b_i = \theta w_s \sum_{i=1}^{n} \bar{w}_i^2 + (1 - \theta)w_o \sum_{i=1}^{n} \bar{w}_i^2, \tag{3.32}$$

where $w_k$ is the comprehensive weight, $k = 1, 2, \ldots, m$.

Based on the mathematical derivation introduced in §4.1, it shows that the hesitation information of experts' evaluation information can be taken into account by using the vague set theory, and then the subjective weights obtained by quantitative analysis are more reasonable. Combined with the objective weight obtained by entropy weight method, the optimization model is established with error minimization to calculate the comprehensive weights. This method can get more accurate and reasonable index weights.

# 4. Application example for underground gas storage

In the following, we take the example of an underground gas storage (in Jiangsu Province, China) for safety evaluation. This gas storage facility has been running for 11 years, and the stratigraphic distribution of the bedded salt deposit and the geometry of the cavity are shown in figure 1. The

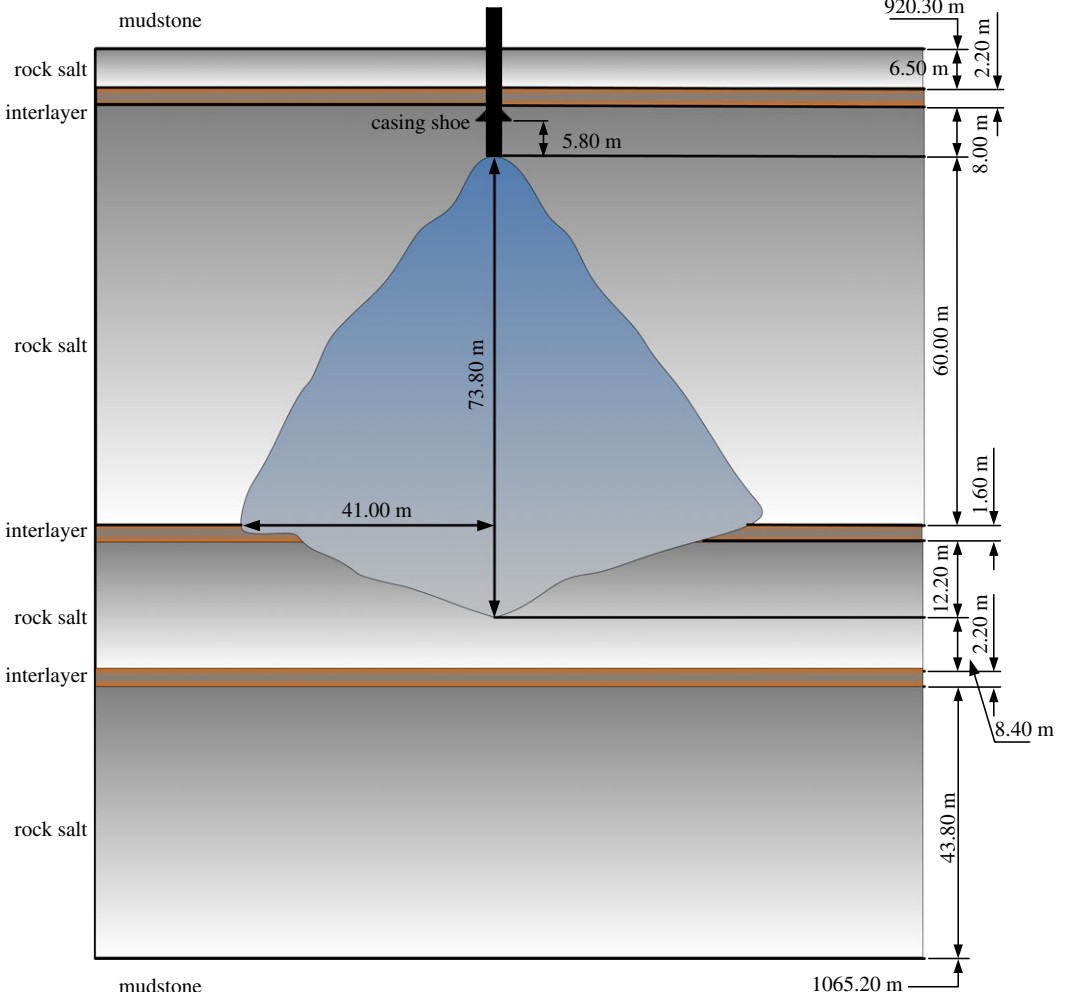

**Figure 1.** Stratigraphic distribution and vertical section of the studied underground gas storage reservoir.

safety indexes values are presented in table 1, considering the units of indexes are not uniform, the values are dimensionless treatment and $R$ represents the ratio of the corresponding index. For sustainable management purposes, the safety of underground gas storage needs to be evaluated.

## 4.1. Safety evaluation index system

To evaluate the safety of gas storage, there should be some evaluation indexes. On the premise of considering the reservoir properties, the external environment and the operation parameters that affect the safety and studying the literature and engineering data, the evaluation indexes are selected and shown in figure 2. In this paper, these indexes are divided into three sets, namely, the salt rock mechanical properties, cavity parameters and operation parameters. Of the three sets, the mechanical properties set contains the indexes that reflect the safety indirectly; for instance, the permeability and creep of salt rock will cause the deformation and capacity reduction of the cavity, but permeability and creep are phenomena generated by the mechanical characteristics of rock salt. The cavity properties contain indexes that describe the storage characteristics, such as the roof thickness, and reflect safety directly. The operation parameters contain the indexes that describe the operation state; for instance, a great change in the operation pressure will accelerate the volume convergence.

In addition to the evaluation indexes, the evaluation criteria for safety also need to be determined, i.e. different levels, from insecurity to security, need to be divided. The purpose of these levels is to characterize the different states of the gas storage and provide references to administrators to achieve the reasonable allocation and utilization of operating resources. By consulting the security levels of relevant literature and practice [27], this paper divides safety into six levels and uses the scoring interval shown in table 2.

**Table 1.** Evaluation indices of safety.

| index | actual value | dimensionless value |
| --- | --- | --- |
| mechanical properties | | |
| elastic modulus (GPa) | 18.50 | 0.93 |
| cohesion (MPa) | 1.20 | 0.24 |
| internal friction angle (℃) | 45.00 | 0.90 |
| steady-state creep rate ($10^{-4}$ $h^{-1}$) | 2.40 | 0.10 |
| cavity parameters | | |
| volume contraction rate (%) | 0.19 | 0.19 |
| the interlayer content (%) | 3 | 0.97 |
| roof thickness ($R$) | 0.26 | 0.53 |
| floor thickness ($R$) | 0.95 | 0.99 |
| height of casing shoe | 0.75 | 0.60 |
| cavity spacing ($R$) | 1.70 | 0.68 |
| stiffness ratio between interbeds and salt rock | 1.62 | 0.81 |
| surrounding rock thickness ($R$) | 2.12 | 0.99 |
| operation parameters | | |
| pressure difference between adjacent cavities ($R$) | 1.75 | 0.22 |
| maximum gas recovery velocity ($R$) | 0.97 | 0.97 |
| minimum pressure ($R$) | 1.09 | 0.91 |
| maximum pressure ($R$) | 0.94 | 0.94 |

## 4.2. Scoring for the evaluation indexes

In this section, we introduce the score curves of the evaluation indexes. The score curves are drawn based on the index properties and storage operation conditions. Additionally, for mathematical analysis of the index, some index values should be pre-processed. The shape of the cavity is represented indirectly by a cubical contraction. The thickness of the roof, bottom plate and surrounding rock are represented indirectly by the ratio of the actual thickness and the cavity diameter. The height of the casing shoe is represented indirectly by the ratio of the actual height and the simulated height. The cavity spacing is represented indirectly by the ratio of the actual spacing and the cavity diameter. The pressure difference between adjacent cavities, the maximum gas production rate, the maximum internal pressure and the minimum internal pressure are represented indirectly by the ratio of the actual value and the simulated value, respectively. Considering that the units of indexes are not uniform, the data are treated as dimensionless.

Figure 3 shows the score curves of the operation parameters. The pressure difference between adjacent cavities must be considered for the salt rock gas storage group. At present, most of the salt rock gas storage reservoirs in the world operate in parallel, that is, several gas storage reservoirs are extracted at the same time, and each cavity has the same pressure at any time. This method was preferred in the Xanten and Epe underground gas storage facilities in Germany. For example, the Xanten underground gas storage extracted the same amount of gas during 90 days in parallel and in series; the total contractive volume of parallel operation was 2.915 m$^3$ less than the series operation. Considering the effect on safety, a greater minimum pressure and smaller maximum gas recovery velocity are better; however, the actual peak load capacity limits the operation pressure and gas recovery velocity to a safe range.

Figure 4 presents the score curves of the interbeds content and stiffness ratio between the interbeds and salt rock. The stiffness of mudstone interbeds is greater than that of the salt rock and can restrain the deformation of gas storage, so the higher the interbeds content is, the more obvious the restraining effect. However, the permeability of mudstone is several orders of magnitude larger than that of salt rock, so the content of mudstone is too high to affect the tightness of gas storage. The stiffness difference between

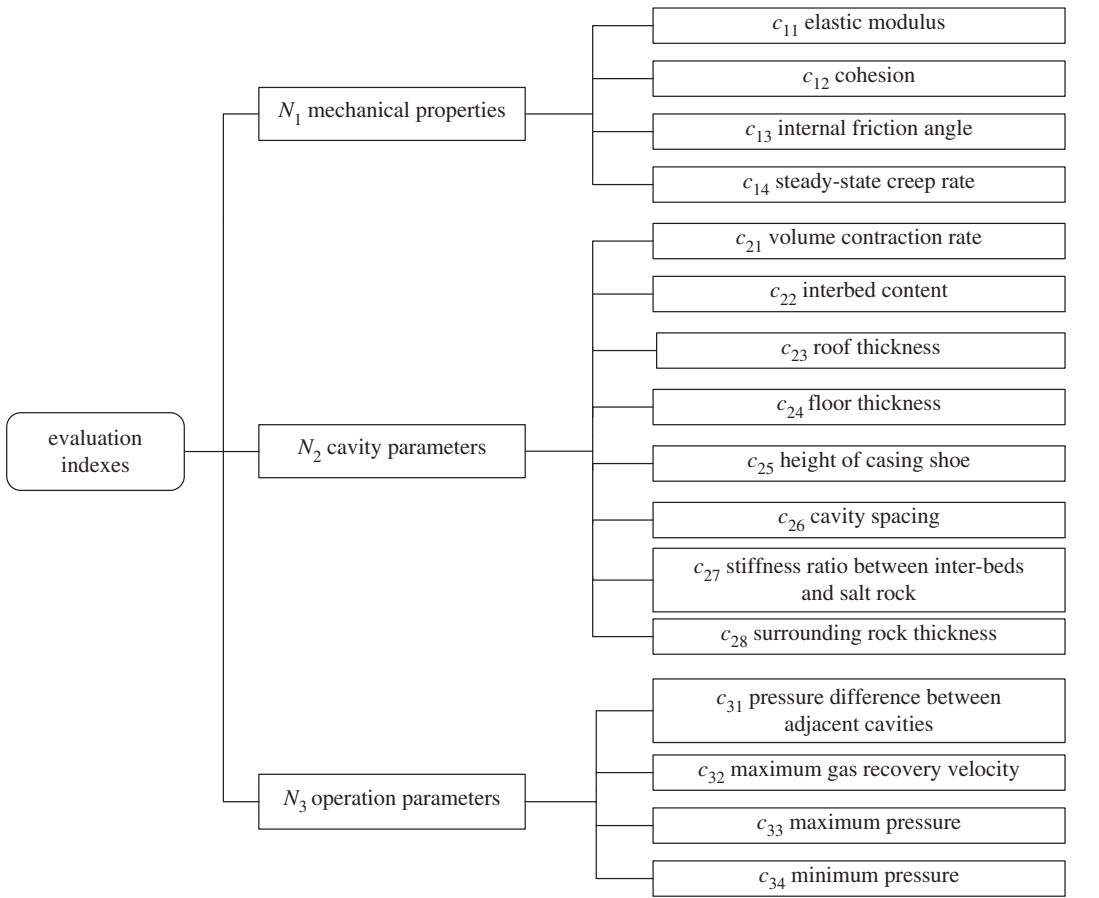

**Figure 2.** Evaluation indexes for the safety of the underground gas storage facility.

mudstone and salt rock is too large to make the deformation of mudstone and salt rock not consistent and accelerate the instability of salt rock gas storage. Therefore, the proportion of interbeds and the ratio of the stiffnesses of the interbeds and salt rock should not be too high.

Figure 5a shows the score curve of the volume contraction rate; the score decreases linearly with the cavity contraction. Figure 5b presents the score curves of the roof thickness. Salt rock has great creep characteristics, and the gas storage with a certain thickness of roof salt rock can offset or weaken the influence of cavern deformation on the upper rock and surface. The ratio of roof thickness to the cavity maximum diameter set by the United States Department of Energy is 10, while the ratio in the Jintan gas storage area is only 3. Most of the salt rocks in China are thin-bedded, and there are no great salt domes for underground energy storage in countries such as Germany and the USA. To ensure the safety of gas storage, the roof, the floor and the surrounding rock of the cavern must be a low-permeability salt rock with sufficient thickness. According to the experience of foreign gas storage construction and the situation of salt resources in China, it is required that the floor and surrounding rock thickness should not be less than the roof thickness. The score curves of the floor thickness and surrounding rock thickness are the same.

Figure 6a shows the score curve for casing shoe height. To prevent the casing from being destroyed by high tensile stress during operation, the casing shoe should be kept a certain distance from the top of the gas storage reservoir. To ensure that the plastic failure of a single cavity does not affect the adjacent cavity, the cavity spacing should be determined by the cavity shape; therefore, according to the ratio of the cavity spacing to the cavity diameter, the score curve of the cavity spacing is shown in figure 6b.

Figure 7a–c presents the score curves for the elastic modulus, cohesion and internal friction angle, respectively. The elastic modulus is the stress needed to determine the elastic deformation in the salt rock under the action of an external force, and the stress required for elastic deformation of rock salt increases with the elastic modulus. The cohesion is the mutual attraction between adjacent parts of the salt rock and reflects the strength of the salt rock; the strength of salt rock increases with cohesion. The internal friction angle refers to the shear strength of salt rock in the process of shear failure, and

**Table 2.** Evaluation levels for safety.

| description | level | scoring interval | definition |
|---|---|---|---|
| disastrous | I | [0, 50) | the influence on the overall safety, tightness and stability is very severe. The storage cavern operations should be cancelled immediately and preparedness measures should be initiated |
| serious | II | [50, 60) | the influence on the overall safety, tightness and stability of the salt cavern is serious. Strict monitoring measures and early warning policies should be formulated |
| marginal | III | [60, 70) | the influence on the overall safety, tightness and stability of the salt cavern should be taken into consideration in the operation process. The monitor of failure parts in the storage cavern should be strengthened |
| improvable | IV | [70, 80) | the influence on the overall safety, tightness and stability of the salt cavern should be taken into consideration in the operation process. The failure parts in the storage cavern should be protected |
| slight | V | [80, 90) | the influence on the overall safety, tightness and stability of the salt cavern can be ignored in the operation process. However, the frequency of regular monitoring should be increased |
| ignorable | VI | [90, 100) | the influence on the overall safety, tightness and stability of the salt cavern can be ignored in the operation process. Only regular monitoring is needed |

the shear strength increases with the internal friction angle. The minimum operating pressure of salt rock gas storage in China is 5–6 MPa, and the overburden pressure is 10–20 MPa. According to the uniaxial compression creep tests and triaxial compression creep tests conducted by Wuhan Institute of Rock and Soil Mechanics [28] on cores (drilled out of the Jintan Salt Mine, Jiangsu Province), the score curve of the steady-state creep rate is shown in figure 7$d$.

Based on the score curves, the index scores under six levels can be calculated, and the dimensionless scores are presented in table 3.

## 4.3. Establish the classical domain, joint domain and evaluation objects

Taking the classical domain of salt rock mechanics as an example, in the classical domain, $U_q$ ($q = 1, 2, \ldots, 6$) is the safety level, and the classical fields $R_1(1)$ to $R_1(6)$ are as follows:

$$R_1(1) = \begin{bmatrix} U_1 & c_{11} & \langle 0, 0.21 \rangle \\ & c_{12} & \langle 0, 0.17 \rangle \\ & c_{13} & \langle 0, 0.55 \rangle \\ & c_{14} & \langle 0.54, 1 \rangle \end{bmatrix}, \quad R_1(2) = \begin{bmatrix} U_2 & c_{11} & \langle 0.21, 0.25 \rangle \\ & c_{12} & \langle 0.17, 0.20 \rangle \\ & c_{13} & \langle 0.55, 0.60 \rangle \\ & c_{14} & \langle 0.50, 0.54 \rangle \end{bmatrix},$$

$$R_1(3) = \begin{bmatrix} U_3 & c_{11} & \langle 0.25, 0.35 \rangle \\ & c_{12} & \langle 0.20, 0.31 \rangle \\ & c_{13} & \langle 0.60, 0.65 \rangle \\ & c_{14} & \langle 0.43, 0.50 \rangle \end{bmatrix}, \quad R_1(4) = \begin{bmatrix} U_4 & c_{11} & \langle 0.35, 0.47 \rangle \\ & c_{12} & \langle 0.31, 0.43 \rangle \\ & c_{13} & \langle 0.65, 0.72 \rangle \\ & c_{14} & \langle 0.35, 0.43 \rangle \end{bmatrix},$$

$$R_1(5) = \begin{bmatrix} U_5 & c_{11} & \langle 0.47, 0.63 \rangle \\ & c_{12} & \langle 0.43, 0.60 \rangle \\ & c_{13} & \langle 0.72, 0.80 \rangle \\ & c_{14} & \langle 0.25, 0.35 \rangle \end{bmatrix} \quad \text{and} \quad R_1(6) = \begin{bmatrix} U_6 & c_{11} & \langle 0.63, 1 \rangle \\ & c_{12} & \langle 0.60, 1 \rangle \\ & c_{13} & \langle 0.80, 1 \rangle \\ & c_{14} & \langle 0, 0.25 \rangle \end{bmatrix}.$$

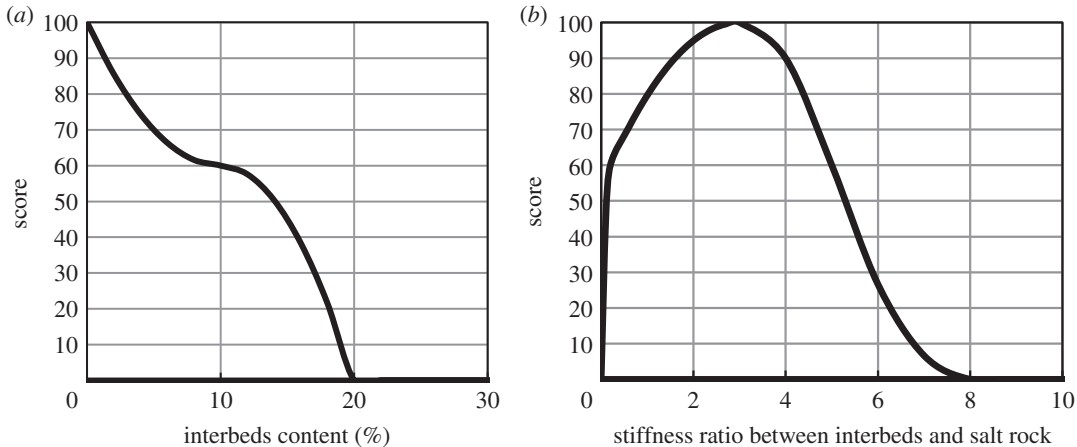

**Figure 3.** Score curves of the operation parameters: (*a*) pressure difference between adjacent cavities, (*b*) maximum pressure, (*c*) maximum gas recovery velocity and (*d*) minimum pressure.

**Figure 4.** (*a*) Score curve of interbed content and (*b*) stiffness ratio between interbeds and salt rock.

For the dimensionless index values, the joint domain $R_u(1)$ is

$$R_u(1) = \begin{bmatrix} U & c_{11} & \langle 0,1 \rangle \\ & c_{12} & \langle 0,1 \rangle \\ & c_{13} & \langle 0,1 \rangle \\ & c_{14} & \langle 0,1 \rangle \end{bmatrix}.$$

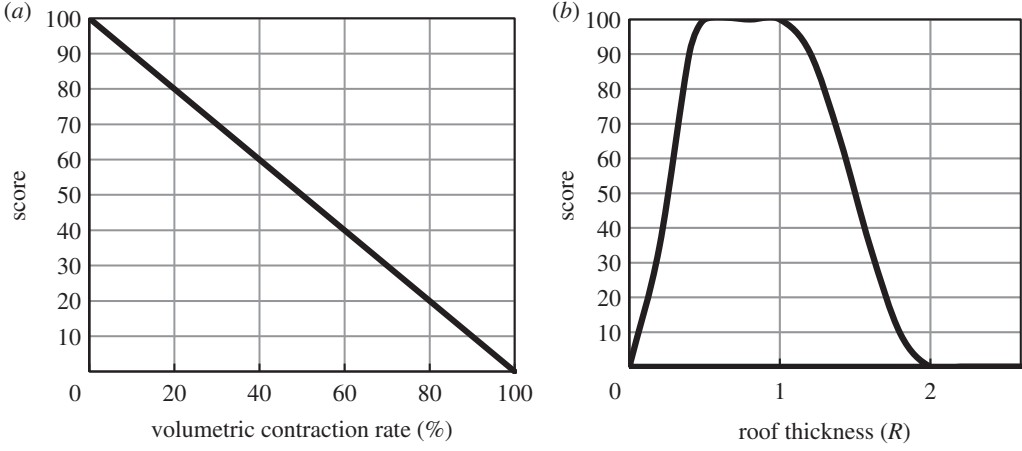

**Figure 5.** Score curve of (*a*) volume contraction rate and (*b*) roof thickness.

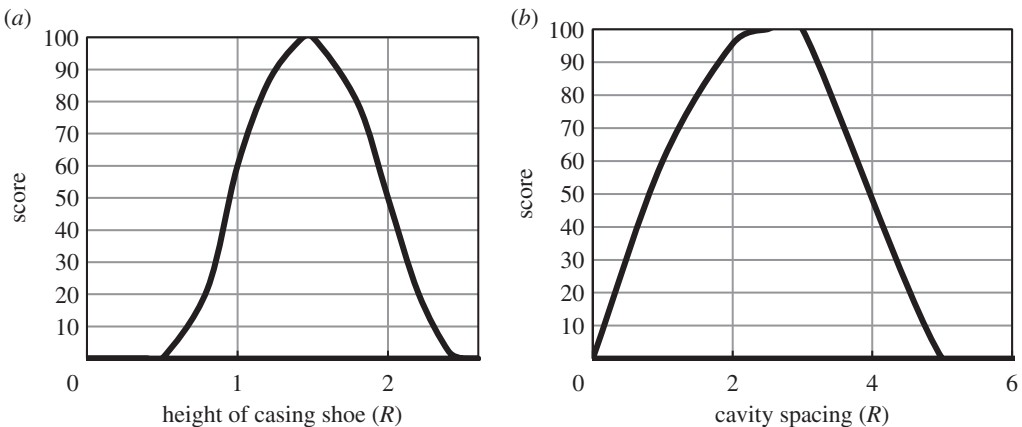

**Figure 6.** Score curve of (*a*) height of casing shoe and (*b*) cavity spacing.

The classical domains and joint domains of the cavity properties and operation parameters follow the same principle. Based on the index values shown in table 1, the matter-elements of the evaluation objects $R_1$ to $R_3$ are

$$
R_1 = \begin{bmatrix} N_1 & c_{11} & 0.93 \\ & c_{12} & 0.24 \\ & c_{13} & 0.90 \\ & c_{14} & 0.10 \end{bmatrix}, \quad
R_2 = \begin{bmatrix} N_2 & c_{21} & 0.81 \\ & c_{22} & 0.97 \\ & c_{23} & 0.25 \\ & c_{24} & 0.95 \\ & c_{25} & 0.60 \\ & c_{26} & 0.68 \\ & c_{27} & 0.81 \\ & c_{28} & 0.99 \end{bmatrix} \quad \text{and} \quad
R_3 = \begin{bmatrix} N_3 & c_{31} & 0.65 \\ & c_{32} & 0.97 \\ & c_{33} & 0.94 \\ & c_{34} & 0.91 \end{bmatrix},
$$

where $N_1$ to $N_3$ are the index sets of the salt rock mechanics, cavity properties and operation parameters, respectively, and $c_{ij}$ represents the indexes.

## 4.4. Calculate the correlation function values of each index

The index correlation values of the salt rock mechanics can be calculated by equations (2.4)–(2.8). Taking the index of the elastic modulus as an example, for the safety level from $V_1$ to $V_6$, the distances of each level are $\rho(v_{11}, V_1) = 0.72$, $\rho(v_{11}, V_2) = 0.68$, $\rho(v_{11}, V_3) = 0.58$, $\rho(v_{11}, V_4) = 0.46$, $\rho(v_{11}, V_5) = 0.30$ and $\rho(v_{11}, V_6) = -0.07$, and the distance of the whole level is $\rho(v_{11}, V_U) = -0.07$. The index correlation value of the elastic modulus is $k_1(c_{11}) = -0.9114$. The correlation values of other indexes follow the same principle, and the calculation results are shown in table 4.

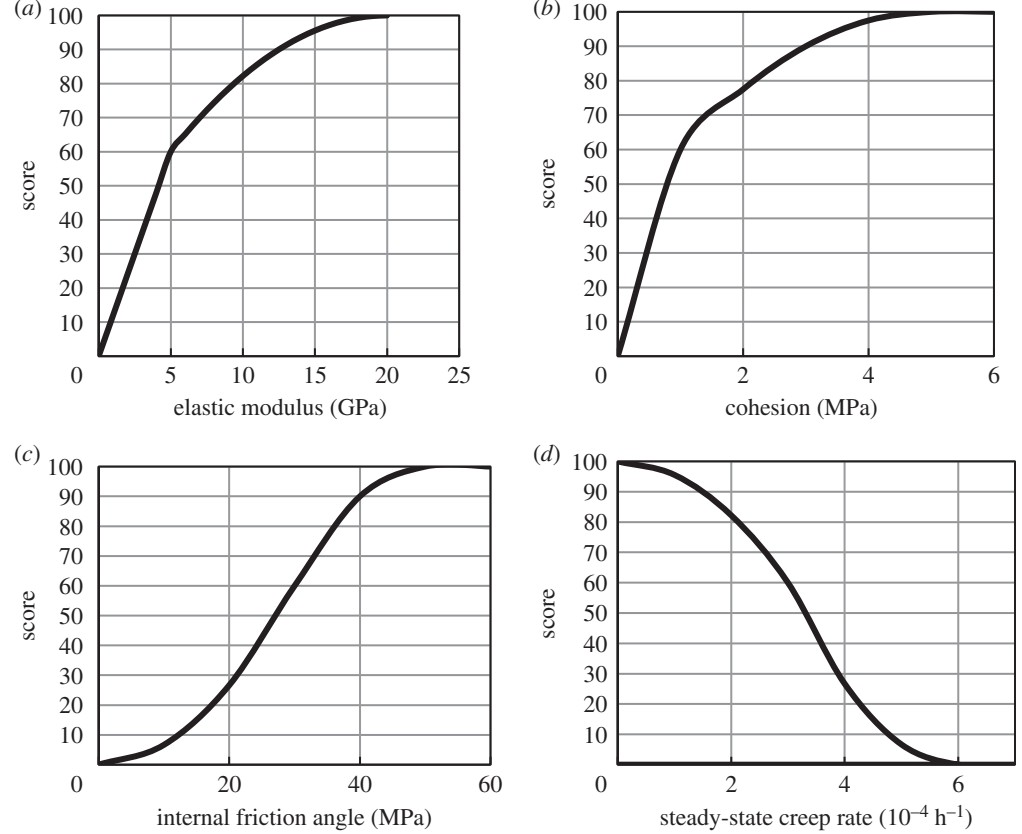

**Figure 7.** Score curves of (*a*) elastic modulus, (*b*) cohesion, (*c*) internal friction angle and (*d*) steady-state creep rate.

**Table 3.** Index values under each level.

| index | I | II | III | IV | V | VI |
|---|---|---|---|---|---|---|
| elastic modulus | 0.00–0.21 | 0.21–0.25 | 0.25–0.35 | 0.35–0.47 | 0.47–0.63 | 0.63–1 |
| cohesion | 0–0.17 | 0.17–0.20 | 0.20–0.31 | 0.31–0.43 | 0.43–0.60 | 0.60–1 |
| internal friction angle | 0–0.55 | 0.55–0.60 | 0.60–0.65 | 0.65–0.72 | 0.72–0.80 | 0.80–1 |
| steady-state creep rate | 0.54–1 | 0.50–0.54 | 0.43–0.50 | 0.35–0.43 | 0.25–0.35 | 0–0.25 |
| volume contraction rate | 0.5–1 | 0.4–0.5 | 0.3–0.4 | 0.2–0.3 | 0.1–0.2 | 0–0.1 |
| interbeds content | 0–0.85 | 0.85–0.90 | 0.90–0.95 | 0.95–0.97 | 0.97–0.99 | 0.99–1 |
| roof thickness | 0–0.50 | 0.50–0.56 | 0.56–0.64 | 0.64–0.70 | 0.70–0.80 | 0.80–1 |
| floor thickness | 0.0–0.60 | 0.60–0.65 | 0.65–0.70 | 0.70–0.72 | 0.72–0.75 | 0.75–1 |
| height of casing shoe | 0–0.20 | 0.20–0.45 | 0.45–0.51 | 0.51–0.56 | 0.56–0.64 | 0.64–1 |
| cavity spacing | 0–0.4 | 0.4–0.45 | 0.45–0.51 | 0.51–0.56 | 0.56–0.76 | 0.76–1 |
| stiffness ratio between interbed sand salt rock | 0–0.04 | 0.04–0.07 | 0.07–0.19 | 0.19–0.34 | 0.34–0.53 | 0.53–1.00 |
| surrounding rock thickness | 0.0–0.60 | 0.60–0.65 | 0.65–0.70 | 0.70–0.72 | 0.72–0.75 | 0.75–1 |
| pressure difference between adjacent cavities | 0.5–1 | 0.44–0.5 | 0.37–0.44 | 0.30–0.37 | 0.20–0.30 | 0–0.20 |
| maximum gas recovery velocity | 0–0.8 | 0.8–0.83 | 0.83–0.85 | 0.85–0.88 | 0.88–0.92 | 0.92–1 |
| minimum pressure | 0–0.75 | 0.75–0.78 | 0.78–0.82 | 0.82–0.85 | 0.85–0.90 | 0.90–1 |
| maximum pressure | 0–0.75 | 0.75–0.78 | 0.78–0.82 | 0.82–0.85 | 0.85–0.90 | 0.90–1 |

**Table 4.** Correlation values of each index.

| indexes | I | II | III | IV | V | VI |
|---|---|---|---|---|---|---|
| elastic modulus | −0.9114 | −0.9067 | −0.8923 | −0.8679 | −0.8108 | −0.9800 |
| cohesion | −0.2258 | −0.1429 | −0.4118 | 3.8000 | 3.0000 | 1.4615 |
| internal friction angle | −0.7778 | −0.7500 | −0.7143 | −0.6429 | −0.5000 | −0.9000 |
| steady-state creep rate | −0.8148 | −0.8000 | −0.7674 | −0.7143 | −0.6000 | −0.9000 |
| volume contraction rate | −0.6200 | −0.5250 | −0.3667 | −0.5000 | 0.0556 | −0.3214 |
| interbeds content | −0.8000 | −0.7000 | −0.4000 | 0.0000 | 0.0000 | −0.4000 |
| roof thickness | −0.0600 | 0.0628 | −0.0600 | −0.1897 | −0.2656 | −0.3649 |
| floor thickness | −0.9750 | −0.9714 | −0.9667 | −0.9643 | −0.9600 | −0.9900 |
| height of casing shoe | −0.5000 | −0.2727 | −0.1837 | −0.0909 | 0.1111 | −0.0909 |
| cavity spacing | −0.4667 | −0.4182 | −0.3469 | −0.2727 | 0.3333 | −0.2000 |
| stiffness ratio between interbeds and salt rock | −0.8021 | −0.7957 | −0.7654 | −0.7286 | 0.4615 | −0.8100 |
| surrounding rock thickness | −0.9750 | −0.9714 | −0.9667 | −0.9643 | −0.9600 | −0.9900 |
| pressure difference between adjacent cavities | 0.7500 | −0.3000 | −0.3750 | −0.4444 | −0.5000 | −0.5625 |
| maximum gas recovery velocity | −0.3617 | −0.3182 | 1.7317 | 1.6522 | 1.5666 | 1.4615 |
| minimum pressure | −0.6400 | −0.5909 | −0.5000 | −0.4000 | −0.1000 | 0.1250 |
| maximum pressure | −0.7600 | −0.7273 | −0.6667 | −0.6000 | −0.4000 | 0.2500 |

**Table 5.** Vague values of the safety levels.

| | disastrous | serious | marginal | improvable | slight | ignorable |
|---|---|---|---|---|---|---|
| vague value | [0.1, 0.2] | [0.2, 0.3] | [0.4, 0.6] | [0.6, 0.8] | [0.8, 0.9] | [0.9, 1] |
| hesitation degree | 0.1 | 0.1 | 0.2 | 0.2 | 0.1 | 0.1 |

## 4.5. Determine the index weight

The subjective weight is determined based on the vague set. According to the six safety levels, the vague values formed by the linguistic variables of the qualitative judgements are shown in table 5.

Ten experts in risk management of underground gas storage were invited to evaluate the safety indexes, and the subjective weights of each index are calculated. Based on the index numbers of each group, the weights of the three index groups are 0.25, 0.5 and 0.25.

The objective weight is determined by the entropy method, and the objective weight values are calculated by equations. (2.9) to (3.1).After obtaining the subjective and objective weights, we obtain the comprehensive weights of each index shown in table 6.

## 4.6. Comprehensive evaluation of underground gas storage

Taking salt rock mechanics as an example, the weights of mechanical parameters $w_1$ = [0.0380, 0.0354, 0.0507, 0.0455] are multiplied by the corresponding index correlation values matrix to obtain the normalized correlation values $K_1$ = [−0.1788, −0.1715, −0.1790, 0.0355, 0.0180, −0.1106] at each security level. The same procedure can be used to obtain the correlation values of the cavity properties and operation parameters. The final calculation results are shown in table 7.

The correlation values are given in equation (2.9) for comprehensive risk calculation, and the eigenvalue of the safety is 4.659. The comprehensive risk of underground gas storage is at the V level.

**Table 6.** Weight values of the indexes.

| index | subjective weight | objective weight | comprehensive weight |
|---|---|---|---|
| elastic modulus | 0.0644 | 0.0116 | 0.0380 |
| cohesion | 0.0485 | 0.0223 | 0.0354 |
| internal friction angle | 0.0707 | 0.0307 | 0.0507 |
| steady-state creep rate | 0.0665 | 0.0245 | 0.0455 |
| volume contraction rate | 0.0646 | 0.1165 | 0.0906 |
| interbeds content | 0.0580 | 0.1226 | 0.0903 |
| roof thickness | 0.0448 | 0.2756 | 0.1602 |
| floor thickness | 0.0638 | 0.0025 | 0.0332 |
| height of casing shoe | 0.0636 | 0.0920 | 0.0778 |
| cavity spacing | 0.0683 | 0.0858 | 0.0771 |
| stiffness ratio between interbeds and salt rock | 0.0679 | 0.0248 | 0.0464 |
| surrounding rock thickness | 0.0689 | 0.0025 | 0.0357 |
| pressure difference between adjacent cavities | 0.0456 | 0.0736 | 0.0596 |
| maximum gas recovery velocity | 0.0697 | 0.0230 | 0.0464 |
| minimum pressure | 0.0688 | 0.0552 | 0.0620 |
| maximum pressure | 0.0659 | 0.0368 | 0.0514 |

**Table 7.** Comprehensive correlation values of the indexes.

| | I | II | III | IV | V | VI |
|---|---|---|---|---|---|---|
| mechanics properties | −0.1191 | −0.1139 | −0.1196 | 0.0364 | 0.0227 | −0.0690 |
| cavity parameters | −0.3172 | −0.2579 | −0.2220 | −0.2039 | −0.0479 | −0.2519 |
| operation parameters | −0.0508 | −0.1066 | −0.0073 | −0.0055 | 0.0161 | 0.0548 |
| comprehensive correlation value | −0.4871 | −0.4784 | −0.3489 | −0.1730 | −0.0091 | −0.2661 |

## 4.7. Discussion

The subjective weighting methods include analytic hierarchy process, Delphi method, binomial coefficient method, fuzzy analysis method, etc. but the key point for safety evaluation of gas storage is how to deal with the fuzziness of the evaluation information. In general, experts cannot give an exact value for the index evaluation; therefore, how to quantify the fuzziness information should be the focus in the design of the subjective weight method; it is better to use fuzzy theory to design the weight method. The objective weighting methods include the entropy method, principal component analysis, Critic method, coefficient of variation method, etc. The central step of these methods is to extract useful information from data samples to reflect the importance of indicators. For gas storage, data samples reflect the objective status of safety, so the key of objective weighting method is to establish a suitable mathematical method to deal with the data to quantify the relationship between indexes. At present, the entropy method is commonly used.

The eigenvalue shows that the security tends to the V level and also that it deviates from the IV level, i.e. the storage could be more stable after improvement. The final calculation results in table 6 show that the safety level of the rock mechanics indexes is IV, the safety level of the cavity properties indexes is V and the safety level of the operation parameters indexes is VI. These results indicate that all the operation parameter indexes are in the healthy state but that the rock mechanics indexes and cavity properties indexes are unstable, and some indexes are defective. The main reason for these results is that the gas storage uses an abandoned salt cavern, and the safety requirements are strict during operation. The salt rock cohesion and the roof thickness are at the II level (i.e. Serious), the volume contraction ratio, the interlayer number, the height of the casing shoe and the adjacent cavity pressure difference are at

the V level (i.e. Slight). Hence, the adjacent cavity pressure difference can be optimized, whereas the other defective indexes are natural properties, which are difficult to improve. However, these defective indexes remain in an acceptable range for storage safety, and passive management measures need to be taken, i.e. with the increase in operating time, the insecurity of these indexes will increase and will need to be monitored in a timely manner.

# 5. Conclusion

In this study, based on the vague set theory and entropy method, the matter-element extension evaluation method was studied and proposed. An evaluation system of the indexes for underground gas storage was established. To further improve the accuracy of evaluation, the index weights were built in the form of comprehensive weights. The subjective weights were obtained by vague set theory, which considered the difference in experts' preferences and played a guiding role in the weights of the indexes, and the objective weights obtained by the entropy method judged the influence according to the values of the indexes, forming the mathematical basis. By minimizing the square error to calculate the comprehensive weights, this method reasonably improved the accuracy of the index weights.

The comprehensive weighting method and matter-element extension method were introduced into the study of safety evaluation of Jintan underground gas storage. The comprehensive weighting method could achieve the weights of single index correlations effectively and rationality. Based on the comprehensive weights, the matter-element extension method was used to evaluate the safety of gas storage and the result was in a safer state. It could be concluded the defective indexes of gas storage include salt rock cohesion, the roof thickness, the volume contraction ratio, the interlayer content, the height of the casing shoe and the adjacent cavity pressure difference.

On the whole, Jintan gas storage has a good safety operation condition, but it still faces the risk of declining security. Therefore, monitoring and adjustments need to be made regularly. At present, only the operation parameters can be controlled. That is to say, with the increase in the operation time of the gas storage, it is necessary to control the maximum gas recovery velocity and ensure the suitability of the operational pressure.

This method can solve the fuzzy problem caused by subjective reasoning in the quantitative process of qualitative judgement for safety evaluation and can be revised by objective data, which provides a reference for the rationale behind sustainable management. The evaluation results can show the actual situation of projects in a quantitative form and prove the validity and practicability of the method. The method can not only be used for safety evaluation but can also find defects in gas storage, which will be beneficial for long-term operation and management for underground gas storage.

Data accessibility. This article has no additional data.
Authors' contributions. A.B. and Y.K. designed the research methodology. A.B. and L.Z. collected and analysed the data. Z.L. guided the writing process and A.B. drafted the manuscript. All authors gave final approval for publication. Z.L. coordinated and secured funding.
Competing interests. We declare we have no competing interests.
Funding. This work was supported by National Natural Science Foundation of China (grant no. 41877527) and Social Science Foundation of Shaanxi Province (grant no. 2018S34).
Acknowledgements. A.B. thanks Z. Xinsheng for helpful discussions. We also thank the editor and anonymous reviewers for their valuable comments and suggestions.

# Appendix A

Basic theory of vague sets

**Definition A.1.** Assuming $U$ is a universe of discourse, and $x$ is an element in $U$, vague set $D$ belonging to $U$ means that there exists a pair of membership degrees $t_D$ and $f_D$ in $U$

$$t_D(x): \to [0,1], f_D(x): \to [0,1], \tag{A 1}$$

with

$$0 \le t_D(x) + f_D(x) \le 1, \tag{A 2}$$

where $t_D(x)$ is the truth-membership degree of $D$ and represents the lower bound of the agreement information that indicates that $x$ belongs to $D$. $f_D(x)$ is the false-membership degree of $D$ and

represents the lower bounds of the disagreement information that indicates that $x$ belongs to $D$. The following uncertainty is introduced

$$\pi_D(x) = 1 - t_D(x) - f_D(x). \tag{A 3}$$

$\pi_D(x)$ is the measurement of unknown information, and a higher value of $\pi_D(x)$ indicates that more unknown information about $x$ belongs to $D$.

When the universe of discourse $U$ is continuous, vague set $A$ can be written as

$$D = \int \frac{[t_D(x), 1 - f_D(x)]}{x}\, dx \tag{A 4}$$

When the universe of discourse $U$ is discrete, vague set $D$ can be written as

$$D = \sum_{i=1}^{n} \frac{[t_D(x_i), 1 - f_D(x_i)]}{x_i} \tag{A 5}$$

**Definition A.2.** Assuming $x$ and $y$ are vague values, $x = [t_D(x), 1 - f_D(x)]$ and $y = [t_D(y), 1 - f_D(y)]$. Let $C_D(x) = t_D(x) - f_D(x)$ and $C_D(y) = t_D(y) - f_D(y)$. The similarity between $x$ and $y$ is expressed as

$$M(x,y) = 1 - \frac{|C_D(x) - C_D(y)|}{2} = 1 - \frac{|t_x - t_y - (f_x - f_y)|}{2} \tag{A 6}$$

where $M(x,y) \in [0,1]$.

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
