## [Reviewer comments · Royal Society Open Science]

Review History

RSOS-191302.R0 (Original submission)

Review form: Reviewer 1 (Giorgio De Tomi)

Is the manuscript scientifically sound in its present form?

Yes

Are the interpretations and conclusions justified by the results?

Yes

Is the language acceptable?

Yes

Do you have any ethical concerns with this paper?

No

Have you any concerns about statistical analyses in this paper?

No

Recommendation?

Accept with minor revision (please list in comments)

Comments to the Author(s)

The paper proposes a an index system for safety evaluation of underground gas storage, with an innovative approach based on the matter-element extension method. A few comments on the current manuscript:

- the evaluation result in the abstract is relevant but it should be better explained, even if briefly, in the abstract. What does that mean? Is it high? Is it sufficient? What value other operations should aim at?
- the objective of the research is explained in the second paragraph of page 3, but it would be important to explain here how the results are going to help other researchers evaluate their experiments replicating the proposed method.
- The explanation of the subjective method of choosing weights in Chapter 4 is somewhat confusing. I recommend the authors re-write the explanation with further details. And perhaps keeping the explanation of the choice of objective weights separated from the subjective weights.
- does the initial statement of section 4.1 come from a reference in the literature?
- I recommend the discussion is expanded to examine alternative scenarios of subjective and objective weights if possible;
- check spelling, for instance, why "Vague" has capital "V" in the conclusion?

Review form: Reviewer 2

Is the manuscript scientifically sound in its present form?

Yes

Are the interpretations and conclusions justified by the results?

Yes

Is the language acceptable?

No

Do you have any ethical concerns with this paper?

No

Have you any concerns about statistical analyses in this paper?

No

Recommendation?

Major revision is needed (please make suggestions in comments)

Comments to the Author(s)

Manuscript Number: RSOS-191302

Title: Comprehensive Weighted Matter-Element Extension Method for the Safety Evaluation of Underground Gas Storage

Journal name: Royal Society Open Science

In this paper, the authors develop a safety evaluation method based on an index system was established and the matter-element extension method. The authors also use a comprehensive weight computation method based on vague sets and entropy for the weight values of each index in the matter-element extension method. Using the proposed safety evolution method, the author computing the safety level of a gas storage facility in the Jintan salt mines as 4.6433. In general, the manuscript has got some potential. The topic discussed in this manuscript is in high interest. the article offers some interesting perspectives and results. However, the manuscript needs some further improved before to be accepted for publication. The reviewer has listed some specific

comments that might be helpful of the authors to further enhance the quality of the manuscript. Please consider the particular comments listed below.

1. Summary

* Please highlight your scientific contribution

2. Introduction

* The Introduction section is too short, some related information is missing. The reviewer believes that following references will enhance the background of the introduction section, please consider citing following papers:

. Evaluating sustainability of water-energy-food (WEF) nexus using an improved matter-element extension model: A case study of China. *Journal of Cleaner Production* 202, 1097-1106, doi:<https://doi.org/10.1016/j.jclepro.2018.08.213> (2018).

*Shale gas industry sustainability assessment based on WSR methodology and fuzzy matter-element extension model: The case study of China. *Journal of Cleaner Production* 226, 336-348, doi:<https://doi.org/10.1016/j.jclepro.2019.03.346> (2019).

* Evaluating water resource sustainability in Beijing, China: Combining PSR model and matter-element extension method. *Journal of Cleaner Production* 206, 171-179, doi:<https://doi.org/10.1016/j.jclepro.2018.09.057> (2019).

3. Matter-element extension evaluation method

4. Design of the comprehensive weighting calculation method

* Detailed description of the source of the data, processing should be considered.

* The method description is too long. Is it better to highlight your improvement of the method and your innovation in methods?

5. Application example for underground gas storage

6. Discussion

The discussion section is two paragraphs. The two paragraphs form a separate section. It is recommended that the sixth section be combined with the fifth section.

7. Conclusion

* I would suggest to write more explicitly the main findings of this study.

Decision letter (RSOS-191302.R0)

21-Jan-2020

Dear Dr Bi,

The editors assigned to your paper ("Comprehensive Weighted Matter-Element Extension Method for the Safety Evaluation of Underground Gas Storage") have now received comments from reviewers. We would like you to revise your paper in accordance with the referee and Associate Editor suggestions which can be found below (not including confidential reports to the Editor). Please note this decision does not guarantee eventual acceptance.

Please submit a copy of your revised paper before 13-Feb-2020. Please note that the revision deadline will expire at 00.00am on this date. If we do not hear from you within this time then it will be assumed that the paper has been withdrawn. In exceptional circumstances, extensions may be possible if agreed with the Editorial Office in advance. We do not allow multiple rounds of revision so we urge you to make every effort to fully address all of the comments at this stage. If deemed necessary by the Editors, your manuscript will be sent back to one or more of the

original reviewers for assessment. If the original reviewers are not available, we may invite new reviewers.

- Data accessibility

If you wish to submit your supporting data or code to Dryad (<http://datadryad.org/>), or modify your current submission to dryad, please use the following link:
<http://datadryad.org/submit?journalID=RSOS&manu=RSOS-191302>

- Competing interests

- Authors' contributions

AB carried out the molecular lab work, participated in data analysis, carried out sequence alignments, participated in the design of the study and drafted the manuscript; CD carried out

the statistical analyses; EF collected field data; GH conceived of the study, designed the study, coordinated the study and helped draft the manuscript. All authors gave final approval for publication.

- Acknowledgements

- Funding statement

on behalf of the Associate Editor, and Professor R. Kerry Rowe (Subject Editor)
openscience@royalsociety.org

Reviewers' Comments to Author:

Reviewer: 1

Comments to the Author(s)

The paper proposes an index system for safety evaluation of underground gas storage, with an innovative approach based on the matter-element extension method. A few comments on the current manuscript:

- the evaluation result in the abstract is relevant but it should be better explained, even if briefly, in the abstract. What does that mean? Is it high? Is it sufficient? What value other operations should aim at?

- the objective of the research is explained in the second paragraph of page 3, but it would be important to explain here how the results are going to help other researchers evaluate their experiments replicating the proposed method.

- The explanation of the subjective method of choosing weights in Chapter 4 is somewhat confusing. I recommend the authors re-write the explanation with further details. And perhaps keeping the explanation of the choice of objective weights separated from the subjective weights.

- does the initial statement of section 4.1 come from a reference in the literature?

- I recommend the discussion is expanded to examine alternative scenarios of subjective and objective weights if possible;

- check spelling, for instance, why "Vague" has capital "V" in the conclusion?

Reviewer: 2
Comments to the Author(s)

Manuscript Number: RSOS-191302

Title: Comprehensive Weighted Matter-Element Extension Method for the Safety Evaluation of Underground Gas Storage

Author's Response to Decision Letter for (RSOS-191302.R0)

See Appendix A.

RSOS-191302.R1 (Revision)

Review form: Reviewer 1 (Giorgio De Tomi)

Is the manuscript scientifically sound in its present form?

Yes

Are the interpretations and conclusions justified by the results?

Yes

Is the language acceptable?

Yes

Do you have any ethical concerns with this paper?

No

Have you any concerns about statistical analyses in this paper?

No

Recommendation?

Accept as is

Comments to the Author(s)

Only minor corrections and adjustments are required along the text, such as spacing between "Table" and "6" in the discussion, and adjusting monitoring with capital "M" in the conclusions. The authors should review the entire manuscript accordingly prior to publication.

Review form: Reviewer 2

Is the manuscript scientifically sound in its present form?

Yes

Are the interpretations and conclusions justified by the results?

Yes

Is the language acceptable?

Yes

Do you have any ethical concerns with this paper?

No

Have you any concerns about statistical analyses in this paper?

No

Recommendation?

Accept as is

Comments to the Author(s)

My concerns from my previous review have been addressed. The authors have addressed most of the concerns I had with the previous review. I would recommend the paper to be accepted for publication

Decision letter (RSOS-191302.R1)

27-Feb-2020

Dear Dr Bi,

It is a pleasure to accept your manuscript entitled "Comprehensive Weighted Matter-Element Extension Method for the Safety Evaluation of Underground Gas Storage" in its current form for publication in Royal Society Open Science. The comments of the reviewer(s) who reviewed your manuscript are included at the foot of this letter.

on behalf of R. Kerry Rowe (Subject Editor)
openscience@royalsociety.org

Reviewer comments to Author:
Reviewer: 2

Comments to the Author(s)

My concerns from my previous review have been addressed. The authors have addressed most of the concerns I had with the previous review. I would recommend the paper to be accepted for publication

Reviewer: 1

Comments to the Author(s)

Only minor corrections and adjustments are required along the text, such as spacing between "Table" and "6" in the discussion, and adjusting monitoring with capital "M" in the conclusions. The authors should review the entire manuscript accordingly prior to publication.

Appendix A

Dear Editor:

Thank you for your kind letter on January 21, 2020. We revised the manuscript ("Comprehensive Weighted Matter-Element Extension Method for the Safety Evaluation of Underground Gas Storage", ID: RSOS-191302) in accordance with the reviewers' comments, and carefully proof-read the manuscript to minimize typographical, grammatical, and bibliographical errors. Here below is our description on revision according to the reviewers' comments.

To Reviewer 1:

1. The reviewer's comment: the evaluation result in the abstract is relevant but it should be better explained, even if briefly, in the abstract. What does that mean? Is it high? Is it sufficient? What value other operations should aim at?

Response: Thanks for the referee's suggestion. We added the explanation for the evaluation result. For the evaluation, there was a reference standard for the evaluation result, which could be used to obtain the safety status for underground the gas storage. We also explained the defect indexes. The detailed revision can be found in Line: 20-27.

2. The reviewer's comment: the objective of the research is explained in the second paragraph of page 3, but it would be important to explain here how the results are going to help other researchers evaluate their experiments replicating the proposed method.

Response: Thanks for the referee's kind advice. We added the main research process and method in the revised manuscript. We also explained the research results that can help researchers make better operation and maintenance decisions. The added contents could allow other researchers to better understand this paper. The detailed revision can be found in Line: 60-68.

3. The reviewer's comment: The explanation of the subjective method of choosing weights in Chapter 4 is somewhat confusing. I recommend the authors re-write the explanation with further details. And perhaps keeping the explanation of the choice of objective weights separated from the subjective weights.

Response: Thanks for the referee's suggestion. We put forward further explanations for the subjective method and wrote the method as 5 steps. At the same time, according to the referee's advice, the explanation of the choice of objective weights was separated from the subjective weight, which makes the content of section 4 more clear. The detailed revision can be found in Line: 119-127.

4. The reviewer's comment: does the initial statement of section 4.1 come from a reference in the literature?

Response: Thanks for the referee's suggestion. The initial statement of section 4.1 was not coming from the reference. It was the summary of engineering experience. For subjective weights, generally, the safety index under an evaluation level cannot be an exact value. For example, the evaluation value in table 5 has value range at each level, because this value is determined by experience given by experts, and it is usually fuzziness.

5. The reviewer's comment: I recommend the discussion is expanded to examine alternative scenarios of subjective and objective weights if possible;

Response: Thanks for the referee's suggestion. The extended discussion of subjective and objective weighting methods was added in section 5.6. We listed some common subjective and objective weighting methods and gave the key points for designing the weighing method. For safety indexes, the key of subjective weighting method is to quantify the fuzziness of evaluation information, and the key of objective weighting method is to make good use of sample data to reflect the relationship between indexes. The detailed revision can be found in first paragraph of section 5.6.

6. The reviewer's comment: check spelling, for instance, why "Vague" has capital "V" in the conclusion?

Response: Thanks for the referee's kind advice. We are very sorry for our incorrect spelling. The spelling has been checked and the errors have been corrected in the revised version.

To Reviewer 2:

1. The reviewer's comment:

Summary

Please highlight your scientific contribution.

Response: Thanks for the referee's suggestion. We added the scientific contribution, the detailed revision can be found in Line: 20-27.

2. The reviewer's comment:

Introduction

The Introduction section is too short, some related information is missing. The reviewer believes that following references will enhance the background of the introduction section, please consider citing following papers:

. Evaluating sustainability of water-energy-food (WEF) nexus using an improved matter-element extension model: A case study of China. *Journal of Cleaner Production* 202, 1097-1106, doi:<https://doi.org/10.1016/j.jclepro.2018.08.213> (2018).

*Shale gas industry sustainability assessment based on WSR methodology and fuzzy matter-element extension model: The case study of China. *Journal of Cleaner Production* 226, 336-348, doi:<https://doi.org/10.1016/j.jclepro.2019.03.346> (2019).

* Evaluating water resource sustainability in Beijing, China: Combining PSR model and matter-element extension method. Journal of Cleaner Production 206, 171-179, doi:<https://doi.org/10.1016/j.jclepro.2018.09.057> (2019).

Response: Thanks for the referee's suggestion. The related references have been added to the paper (Reference number:24,25,26), but we think the three references are more suitable to add the section 3, because section 3 is used to introduce the matter-element extension method, the references can enhance the background of this method , the detailed revision can be found in Line:79.

3 The reviewer's comment:

Design of the comprehensive weighting calculation method

Detailed description of the source of the data, processing should be considered.

The method description is too long. Is it better to highlight your improvement of the method and your innovation in methods?

Response: Thanks for the referee's good evaluation and kind suggestion. The safety indexes data of the underground gas storage have been added to the table 1, and the detailed data processing is shown in the first paragraph of section 5.2. Section 4 was only described the proposed method steps, but the design steps of subjective weighting method are the innovation actually, now the innovations are further explained in revised manuscript and the detailed revision can be found in last paragraph of section 4 and first paragraph in section 5.

4. The reviewer's comment:

Discussion

The discussion section is two paragraphs. The two paragraphs form a separate section. It is recommended that the sixth section be combined with the fifth section.

Response: Thanks for the referee's advice. The two paragraphs of the discussion section are formed into one paragraph, and the section 6 has been combined as a discussion for the examples with section 5, the detailed revision can be found in section 5.6.

5. The reviewer's comment:

Conclusion

I would suggest to write more explicitly the main findings of this study.

Response: Thanks for the referee's suggestion. In the conclusion part, we added the main findings of the study, suggestions for maintenance and operation are also given. The detailed revision can be found in Line: 377-387.

We tried our best to improve the manuscript and all the lines and pages indicated above are in the revised manuscript. Thank you and all the reviewers for the kind advice.

We appreciate for Editors/Reviewers' warm work earnestly, and hope that the correction will meet with approval.

Once again, thank you very much for your comments and suggestions.

Sincerely yours

Aorui Bi, Zhengshan Luo, Yulei Kong, Lexin Zhao.

Corresponding author: Aorui Bi

E-mail: bar_wayne@xauat.edu.cn